# Prognostic Value of Dual-Time-Point ^18^F-Fluorodeoxyglucose PET/CT in Metastatic Breast Cancer: An Exploratory Study of Quantitative Measures

**DOI:** 10.3390/diagnostics10060398

**Published:** 2020-06-11

**Authors:** Mohammad Naghavi-Behzad, Charlotte Bjerg Petersen, Marianne Vogsen, Poul-Erik Braad, Malene Grubbe Hildebrandt, Oke Gerke

**Affiliations:** 1Department of Clinical Research, University of Southern Denmark, 5000 Odense, Denmark; charlottebjergp@gmail.com (C.B.P.); Marianne.Vogsen@rsyd.dk (M.V.); Malene.Grubbe.Hildebrandt@rsyd.dk (M.G.H.); Oke.Gerke@rsyd.dk (O.G.); 2Department of Nuclear Medicine, Odense University Hospital, 5000 Odense, Denmark; poul-erik.braad@rsyd.dk; 3Open Patient data Explorative Network (OPEN), Odense University Hospital, 5000 Odense, Denmark; 4Department of Oncology, Odense University Hospital, 5000 Odense, Denmark; 5Centre for Innovative Medical Technology, Odense University Hospital, 5000 Odense, Denmark; 6Centre for Personalized Response Monitoring in Oncology, Odense University Hospital, 5000 Odense, Denmark

**Keywords:** metastatic breast cancer, FDG-PET/CT, metabolic tumor volume, prognostic value, total lesion glycolysis

## Abstract

This study aimed to compare the prognostic value of quantitative measures of [^18^F]-fluorodeoxyglucose positron emission tomography with integrated computed tomography (FDG-PET/CT) for the response monitoring of patients with metastatic breast cancer (MBC). In this prospective study, 22 patients with biopsy-verified MBC diagnosed between 2011 and 2014 at Odense University Hospital (Denmark) were followed up until 2019. A dual-time-point FDG-PET/CT scan protocol (1 and 3 h) was applied at baseline, when MBC was diagnosed. Baseline characteristics and quantitative measures of maximum standardized uptake value (SUVmax), mean standardized uptake value (SUVmean), corrected SUVmean (cSUVmean), metabolic tumor volume (MTV), total lesion glycolysis (TLG), and corrected TLG (cTLG) were collected. Survival time was analyzed using the Kaplan–Meier method and was regressed on MTV, TLG, and cTLG while adjusting for clinicopathological characteristics. Among the 22 patients included (median age: 59.5 years), 21 patients (95%) died within the follow-up period. Median survival time was 29.13 months (95% Confidence interval: 20.4–40 months). Multivariable Cox proportional hazards regression analyses of survival time showed no influence from the SUVmean, cSUVmean, or SUVmax, while increased values of MTV, TLG, and cTLG were significantly associated with slightly higher risk, with hazard ratios ranging between 1.0003 and 1.004 (*p* = 0.007 to *p* = 0.026). Changes from 1 to 3 h were insignificant for all PET measures in the regression model. In conclusion, MTV and TLG are potential prognostic markers for overall survival in MBC patients.

## 1. Introduction

Breast cancer is the most commonly diagnosed cancer in women and one of the most frequent causes of cancer death worldwide [1]. Overall 10-year survival for patients diagnosed with metastasis is around 10% [2].

Various imaging modalities have been suggested for diagnosing recurrent breast cancer, among which ^18^F-fluorodeoxyglucose positron emission tomography (FDG-PET) with integrated computed tomography (FDG-PET/CT) is more accurate when compared to conventional imaging techniques [3]. FDG-PET/CT is even more accurate compared to conventional imaging in the detection of bone metastases (bone being one of the most probable sites of metastases in breast cancer) [4]. FDG-PET/CT has also proved superior to conventional imaging for predicting treatment response in patients treated for metastatic breast cancer (MBC) [5,6].

Semiquantitative methods based on the changes in the metrics for FDG-PET/CT have been recommended to improve reproducibility in response monitoring for MBC patients [7]. Several studies have aimed to investigate the prognostic value of FDG-PET/CT for MBC patients, often based on the quantification of the maximum standardized uptake value (SUVmax) [8,9,10], although this measure might not be optimal to assess breast cancer since some of the malignant lesions have low FDG uptake [11,12]. However, SUVmax is based only on the one voxel with the highest FDG uptake and may not be optimal for assessing systemic malignant diseases. Therefore, quantification parameters such as total lesion glycolysis (TLG) and metabolic tumor volume (MTV), which represent all malignant lesions, are considered to improve the estimation of total tumor burden [13,14].

FDG-PET/CT has been indicated as an independent predictor of patients’ overall survival even after adjusting for prognostic factors [15], although there is considerable controversy among previous studies regarding an optimal PET measure. This study aimed to compare the prognostic value, in terms of overall survival, of various quantitative PET measurements in patients with metastatic breast cancer.

## 2. Materials and Methods

### 2.1. Study Design and Subjects

This exploratory study was conducted at the Department of Nuclear Medicine at Odense University Hospital (Odense, Denmark) and is a sub-study of a prospective diagnostic study at our research unit [3]. In brief, women with a history of breast cancer and who were referred to our department between 2011 (October) and 2014 (September) on suspicion of a recurrence of breast cancer were evaluated in order to be enrolled. Inclusion criteria comprised female patients over 18 years of age, biopsy-verified MBC, available medical oncological records, and patients who had been evaluated by dual-time-point FDG-PET/CT within 14 days of diagnosis for metastasis. Exclusion criteria were a weight of over 90 kg or under 50 kg, periods of pregnancy and breastfeeding during the study, a blood glucose level of more than 8.0 mmol/L, a history of other malignancy or diabetes mellitus, emigration or lost to follow-up, and mortality due to reasons other than MBC. Among 102 patients with a suspected recurrence of breast cancer, 22 patients with a diagnosis of MBC met our inclusion criteria and were asked to undergo dual-time-point FDG-PET/CT (1 and 3 h) at baseline. The study protocol was approved by the Danish Patient Safety Authority and the Danish Data Protection Agency. Permission was given from the local ethics committee (S-20110138), and informed consent was obtained from all included patients. The study was registered at ClinicalTrials.gov (NCT01552655) in February 2012.

Data regarding patients’ age, time until relapse, type of surgery for their primary tumor, tumor histopathology, site of metastases, grade of malignancy, immunohistochemistry profile (estrogen, progesterone, and herceptin-2 receptors’ status), Ki-67 proliferation, and date of last clinical visit or death were extracted. Follow-up time was defined as the time interval between the date of the baseline FDG-PET/CT scan and the dates of the last registered clinical visit and death for survivors and non-survivors, respectively. Patients were followed up until 2019 (October) for survival analysis. Quantitative variables for the dual-time-point FDG-PET/CT scans consisted of SUVmax, mean standardized uptake value (SUVmean), corrected SUVmean (cSUVmean), MTV, TLG, and corrected TLG (cTLG).

### 2.2. FDG-PET/CT Protocol

Before the FDG-PET/CT scan, patients were required to fast for at least 6 h, after which their blood sugar levels were measured. PET/CT was considered acceptable at levels up to 144 mg/dL. The 18F-FDG tracer was administered intravenously with an activity of 4 MBq per kg of body weight. The patients were requested to rest for 60 min (±5 min) p.i. before PET/CT imaging was performed from the base of the skull to the proximal femur. The second scan was performed in the same manner after 180 min (±5 min). The total examination time was approximately 210 min for each patient.

The scans were performed using either the Discovery STE (VCT) equipped with BGO crystals or the Discovery RX equipped with LYSO(Ce) crystals (GE Healthcare Systems, Chicago, IL, USA). Low-dose CT, with two scout views for both exams, was initially acquired at 140 kVp with SmartmA tube current modulation (Noise Index: 25; 30–110 mA) and used for PET attenuation correction. PET was performed over 7–9 bed positions in 3D, with a scan time of 2.5 min per bed position for 1-h images and 3.5 min per bed position for 3-h images. PET images were reconstructed iteratively, with ordered subset expectation maximization, 2 iterations, and 21 or 28 subsets.

### 2.3. Reference Standard and Quantitative Variables

A biopsy with histopathological examination was used to verify recurrent disease and served as a reference standard. Experienced nuclear medical physicians determined sites of metastases using a visual assessment of the 1 and 3 h FDG-PET/CT scans. The physicians’ assessments served as the basis for the quantitative analyses. The semiquantitative analyses of both the 1 and 3 h FDG-PET/CT scans were performed with the semiautomatic quantification software ROVER (ABX, Radeberg, Germany). Masks were placed around all regions of interest (ROIs), and lesions were measured with an iterative algorithm with a threshold of 40% of maximum.

SUVmax was defined as the maximum uptake in the volume of interest (VOI) that reflects the maximum tissue concentration of FDG uptake in the tumor. SUVmean was taken as the average of the SUV values contained in the VOI. To compute SUVmax, all possible averages for SUV in cubes of 3 × 3 × 3 voxels included in the tumor were computed, and the lesion with the highest uptake of 18F-FDG was selected as the target lesion for SUVmax analysis. In the following step, partial volume correction was performed by ROVER and by using a predefined algorithm for the spilled-out lesion of the ROI, resulting in cSUVmean. MTV was the volume of the VOI after the segmentation. TLG was taken as the product of SUVmean by MTV, providing an estimate of the total tumor glycolytic activity [16].

Patient-based MTV, SUVmean, and cSUVmean were calculated automatically by the quantification software based on all lesions measured in each patient, with the exclusion of two cerebral lesions in two patients, which could not be assessed by the quantification software. TLG with or without correction for the partial volume effect was calculated for each patient using the following equations: (1)TLG=MTV×SUVmean or cTLG=MTV×cSUVmean

### 2.4. Statistical Analyses

Descriptive statistics were performed according to data type (continuous: median and range; categorical: frequencies and percentages). A Kaplan–Meier survival curve including pointwise 95% confidence intervals (CI) was created for visualization purposes. It represents the time interval between the date of metastatic confirmation (the starting point of the metastatic disease) and the date of death or the last clinical visit. Cox proportional hazards regression of FDG-PET/CT measures of age, cancer type, and site of metastases were performed for overall survival for exploratory purposes. The most clinically relevant explanatory variables were chosen, and the number of these was restricted to three due to a total sample size of 22 (with 21 events). For FDG-PET/CT measures, both absolute values and changes from 1 to 3 h were investigated. The latter comprised both absolute and relative changes, including regression of the 3-h measurement on the 1-h measurement and other covariates. *p*-values of <0.05 were considered significant. All statistical analyses were conducted with STATA/MP 16 (StataCorp, College Station, 77845 TX, USA).

## 3. Results

### 3.1. Demographic Information

Twenty-two patients with a median age of 59.5 years (range: 37–76 years) were followed up. The median time period between primary diagnosis and secondary relapse was 58 months (range: 11–324 months), and 15 patients (68.2%) had received adjuvant chemotherapy before. The baseline characteristics of the included patients are summarized in Table 1.

### 3.2. Clinical Information and FDG-PET Measures

Among the study subjects, 21 patients (95%) died within the follow-up period. The median survival time was 29.1 months (95% CI: 20.4–40.0 months) (Figure 1). The median time between the FDG-PET/CT scan and the last follow-up was 29.4 months (range: 0.96–72.3 months). Quantitative FDG-PET measures after 1 and 3 h are summarized in Table 2. Delayed imaging implicated lower tumor volume and increased FDG avidity.

### 3.3. Cox Proportional Hazards Analyses

In multivariable Cox proportional hazards regression analyses, SUVmean (1 and 3 h), cSUVmean (1 and 3 h), and SUVmax (1 and 3 h) were not prognostic for survival, while MTV (1 and 3 h), TLG (1 and 3 h), and cTLG (1 and 3 h) were all prognostic for survival, with hazard ratios ranging from 1.0003 to 1.004 (*p* = 0.007 to *p* = 0.026), as indicated in Table 3. For instance, a hazard ratio of 1.004 for MTV (1 h) translates, on average, into an increase of 4% in risk of death per 10 units of MTV. Likewise, a hazard ratio of 1.0008 for TLG (1 h) translates, on average, into an increase of 8% in the risk of death per 100 units of TLG. Having a soft tissue metastasis was associated with hazard ratios of 10.6–12.5 (*p* = 0.008 to *p* = 0.011) in all six regression analyses, although the respective 95% confidence intervals were wide due to the limited sample size. Both absolute and relative changes from 1 to 3 h were statistically insignificant predictors for overall survival for all PET measures in multivariable Cox proportional hazards regression.

## 4. Discussion

The results of this study indicated that MTV, TLG, and cTLG with both 1- and 3-h scans have prognostic value for overall survival in metastatic breast cancer. However, increased risks were notable only for comparably large increases in MTV (10 units) as well as TLG and cTLG (100 units). In contrast, SUVmean, cSUVmean, and SUVmax were not prognostic for overall survival, nor were absolute and relative changes for 1 to 3 h for all six parameters.

The strengths of this study consisted of prospective data collection and long-term follow-up of the included patients. Additionally, the diagnosis of recurrence and the assessment of metastatic lesions were confirmed by histopathological biopsy as the gold standard. Furthermore, the scans were conducted based on a dual-time-point protocol, and the semiquantitative PET measurements were obtained for both early and delayed imaging. The main limitation of this study was a low sample size, which prohibits drawing firm conclusions.

MTV and TLG (both 1- and 3-h scans) were both independent predictors of patients’ survival. These findings are in line with a similar study (comprising 40 patients) in which pretreatment MTV was found to be an independent prognostic indicator of patients’ overall survival, while SUVmax did not show any association [17]. In another study with 47 triple-negative MBC patients, it has been shown that MTV could act as an independent strong prognostic factor, while SUVmax and TLG were not statistically significant predictors of patients’ overall survival [14].

However, SUVmax was found to be associated with patients’ overall survival in other studies on breast cancer patients with distant metastasis [8,9,18,19]. It has also been reported that MTV and TLG were not predictive of patients’ overall survival [18], which stands in contrast to our results. One reason could be the long follow-up time of our study (approximately 29 months) compared to previous studies with follow-up times of 15.6 [18], 26.6 [8], and 22.3 months [9].

The results of our study were derived from a dual-time-point protocol for which there was no statistically significant difference in patient-based accuracy results between the 1- and 3-h FDG-PET/CT scans [3]. Some potential advantages of using dual-time-point FDG-PET/CT have previously been reported [20], but increased diagnostic accuracy resulting from the implementation of this protocol was not proved [3,21].

Our results indicated that MTV and TLG are the most promising prognostic markers for overall survival in patients with metastatic breast cancer. However, there is great controversy in the literature regarding the selection of an optimal PET measure with high prognostic value. Further research in larger samples is warranted before an optimal quantitative measure can be recommended.

## 5. Conclusions

In the investigation of different PET measures, MTV and TLG appeared to have high prognostic value in metastatic breast cancer patients in terms of overall survival. Both parameters, which could be obtained from routine whole-body FDG-PET/CT, were found to be independent prognostic factors in a small multivariable model, despite a limited sample size (22 patients with 21 events).

## Figures and Tables

**Figure 1 diagnostics-10-00398-f001:**
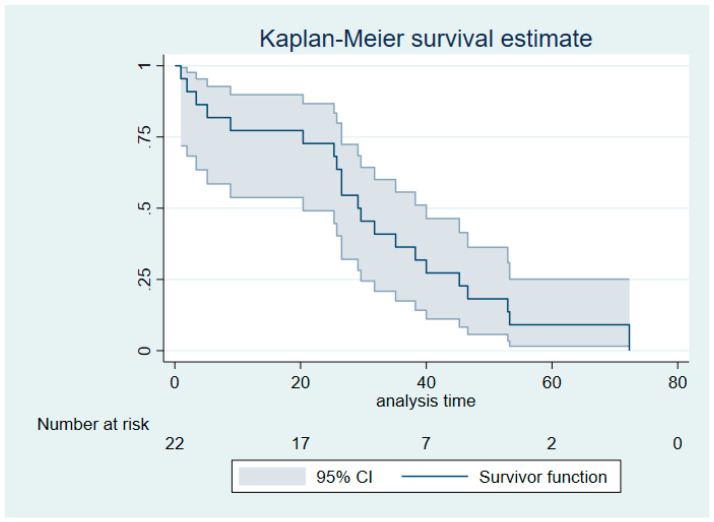
Kaplan–Meier plot for the cohort, including pointwise 95% confidence intervals (analysis time in months).

**Table 1 diagnostics-10-00398-t001:** Baseline characteristics of included patients with recurrent breast cancer.

Variable	Frequency (%)
Cancer type	
Invasive ductal carcinoma	18 (82)
Invasive lobular carcinoma	4 (18)
Type of surgery	
Lumpectomy	11 (50)
Mastectomy	11 (50)
Site of metastases	
Bone	7 (32)
Soft tissue	4 (18)
Bone and soft tissue	11 (50)
Ki-67 proliferation	
0–10%	0 (0)
10–20%	2 (9)
>20%	5 (22.5)
Missing	15 (68.2)
Estrogen receptor status	
Positive	17 (77.3)
Negative	4 (18.2)
Missing	1 (4.5)
Progesterone receptor status	
Positive	8 (36.4)
Negative	13 (59.1)
Missing	1 (4.5)
Herceptin-2 receptor status	
Positive	3 (13.6)
Negative	18 (81.8)
Missing	1 (4.5)
Malignancy Grade	
1	3 (13.6)
2	9 (40.9)
3	8 (36.4)
Missing	2 (9)

**Table 2 diagnostics-10-00398-t002:** Median (range) of the [^18^F]-fluorodeoxyglucose positron emission tomography measures.

Variable	Time Point	
	1 h	3 h
SUVmean	3.89 (1.6–8.4)	5.00 (2.4–9.8)
SUVmax	9.60 (4.2–19.7)	12.45 (5.8–29.7)
MTV	51.15 (4.4–744.7)	42.00 (4.8–747.7)
TLG	201.60 (20.4–4545.5)	216.70 (21.1–4908.1)
cTLG	295.80 (32.9–6219.9)	376. 60 (39.8–7999.9)
cSUVmean	6.94 (2.3–15.0)	8.51 (4.3–20.5)

SUV: standardized uptake value; MTV: metabolic tumor volume; TLG: total lesion glycolysis; cTLG: corrected TLG; cSUVmean: corrected SUVmean.

**Table 3 diagnostics-10-00398-t003:** Multiple Cox regression analyses of the PET measures on baseline characteristics.

Variable	HR	95% CI	*p*	Variable	HR	95% CI	*p*
MTV 1 h	1.004	1.001–1.007	0.007	MTV 3 h	1.004	1.001–1.007	0.01
Age	1.05	0.99–1.11	0.09	Age	1.05	0.99–1.11	0.09
Cancer type				Cancer type			
IDC	Reference			IDC	Reference		
ILC	0.39	0.09–1.59	0.19	ILC	0.38	0.09–1.53	0.17
Site of metastases				Site of metastases			
Bone	Reference			Bone	Reference		
Bone and soft tissue	3.31	0.95–11.50	0.06	Bone and soft tissue	3.44	0.998–11.87	0.05
Soft tissue	12.49	1.91–81.63	0.008	Soft tissue	12.11	1.88–78.09	0.009
TLG 1 h	1.0008	1.0002–1.001	0.009	TLG 3 h	1.0005	1.0001–1.001	0.016
Age	1.05	0.99–1.11	0.08	Age	1.05	0.99–1.11	0.08
Cancer type				Cancer type			
IDC	Reference			IDC	Reference		
ILC	0.45	0.11–1.85	0.27	ILC	0.44	0.11–1.81	0.26
Site of metastases				Site of metastases			
Bone	Reference			Bone	Reference		
Bone and soft tissue	3.28	0.95–11.41	0.06	Bone and soft tissue	3.27	0.94–11.35	0.06
Soft tissue	12.03	1.87–77.25	0.009	Soft tissue	11.25	1.79–70.70	0.01
cTLG 1 h	1.0004	1.0001–1.0008	0.015	cTLG 3 h	1.0003	1.00004–1.0006	0.026
Age	1.05	0.99–1.11	0.08	Age	1.05	0.99–1.11	0.08
Cancer type				Cancer type			
IDC	Reference			IDC	Reference		
ILC	0.44	0.11–1.80	0.25	ILC	0.43	0.11–1.76	0.24
Site of metastases				Site of metastases			
Bone	Reference			Bone	Reference		
Bone and soft tissue	3.37	0.98–11.62	0.055	Bone and soft tissue	3.35	0.97–11.56	0.056
Soft tissue	11.36	1.79–72	0.01	Soft tissue	10.61	1.71–65.74	0.011

MTV: metabolic tumor volume; TLG: total lesion glycolysis; cTLG: corrected TLG; IDC: Invasive ductal carcinoma; ILC: Invasive lobular carcinoma.

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
