# Peer review of "Prognostic Value of Dual-Time-Point 18F-Fluorodeoxyglucose PET/CT in Metastatic Breast Cancer: An Exploratory Study of Quantitative Measures"

_diagnostics, 2020, doi:10.3390/diagnostics10060398_

Round 1

Reviewer 1 Report

ID diagnostics-831412

"Prognostic Value of Dual-time-point 18F-Fluorodeoxyglucose PET/CT in Metastatic Breast Cancer: An Exploratory Study of Quantitative Measures" to be published in Diagnostics.

General remarks

The authors presented a study which aimed to compare the prognostic value of quantitative measures of [18F]-Fluorodeoxyglucose-Positron Emission Tomography with integrated computed-tomography (FDG-PET/CT) for response monitoring of patients with metastatic breast cancer (MBC). 22 patients with biopsy-verified MBC diagnosed between 2011-2014 at Odense University Hospital (Denmark) were followed up until 2019. A dual-time-point FDG-PET/CT scan protocol (1h and 3h) was applied at baseline, when MBC was diagnosed. Baseline characteristics and quantitative measures of SUVmax, SUVmean, corrected-SUVmean (cSUVmean), metabolic tumor volume (MTV), total lesion glycolysis (TLG), and corrected-TLG (cTLG) were collected. Survival time was analyzed by Kaplan-Meier method and was regressed on MTV, TLG, and cTLG adjusting for clinicopathological characteristics. Among included 22 patients, 21 patients (95%) died within the follow-up period. Median survival time was 29.13 months. Multivariable Cox proportional-hazards regression analyses of survival time showed no influence of SUVmean, cSUVmean, and SUVmax, while increased values of MTV, TLG, and cTLG were significantly associated with slightly higher risk with hazard ratios ranging between 1.0003-1.004. Changes from 1h to 3h were insignificant for all PET measures in regression model. Authors concluded that MTV and TLG are potential prognostic markers for overall-survival in MBC patients.

The topic of the article is very interesting and it could serve as an important basis for further research. Namely, breast cancer is the most commonly diagnosed cancer in women and there are various imaging modalities for diagnosing its recurrence.

There is only one minor correction needed before the article is suitable for publication, marked in the attached manuscript.

Author Response

Point 1: There is only one minor correction needed before the article is suitable for publication

Response 1: Thanks for the reviewer’s consideration. The mentioned point has been corrected in revised manuscript (Table 2, line 158, page 5).

Reviewer 2 Report

I appreciate the study design and the clarity of the research.

I would suggest to underline that the number of patients is limited and this should be stated as a limit of the study.

Regarding the data maybe the detailed formula of SUVmax, TLG, MTV etc, beyond their definitions, exceed the readers interests and should be limited.

I have the question if for the dual PET/CT study you used one single scout CT or you performed 2 for each study at 1 h and  3h; if is so, you should write; even if it is a low CT, the irradiation counts.

Author Response

Point 1: I would suggest to underline that the number of patients is limited and this should be stated as a limit of the study.

Response 1: Thank you for the relevant consideration. We fully agree in this point, and it has therefore already been addressed as a limitation in the Discussion section (Discussion section, line 189, page 6).

Point 2: Regarding the data maybe the detailed formula of SUVmax, TLG, MTV etc, beyond their definitions, exceed the readers interests and should be limited.

Response 2: Thank you for the comment, which we also partly agree in.  However, the main purpose of this study was comparison of different PET quantitative measures, and we believe it should be transparent for the readers how exactly we calculated each measure. Therefore, we prefer to keep those definitions in our Methods section, but we limited the definitions of measures in the description part of Table 2 (revised manuscript).

Point 3: I have the question if for the dual PET/CT study you used one single scout CT or you performed 2 for each study at 1 h and  3h; if is so, you should write; even if it is a low CT, the irradiation counts.

Response 3: Thank you very much for the relevant question. Two scouts were performed for both exams in our study. Although, one extra scout gives double the dose than two scouts, the increase in total exam dose is ignorable in worst case. In the end, the extra scout facilitates better tube current modulation in the helical CT, which may actually lead to better image quality as well as lower total exam dose. We added this information to the FDG-PET/CT Protocol section (Line 102, page 3).